# Valuable Data “Gain” and “Loss”: The Quantitative Impact of Information Choice on Consumers’ Decision to Buy Selenium-Rich Agricultural Products

**DOI:** 10.3390/foods13203256

**Published:** 2024-10-13

**Authors:** Bo Zhou, Huizhen Wu, Baoshu Wu, Zhenjiang Song

**Affiliations:** 1College of Economics and Management, Jiangxi Agricultural University, Nanchang 330045, China; zhoubo@jxau.edu.cn (B.Z.); 1939594050@stu.jxau.edu.cn (H.W.); 2School of Business Administration, Jiangxi University of Finance and Economics, Nanchang 330032, China; 3Rural Development Research Center of Jiangxi Province, Jiangxi Agricultural University, Nanchang 330045, China; 4Institute of Jiangxi Selenium-Rich Agricultural Research, Jiangxi Agricultural University, Nanchang 330045, China

**Keywords:** nutritional information intervention, hidden hunger, consumer behavior, food consumption, health food market, selenium-rich agricultural products, purchase decisions

## Abstract

Biotechnology assumes a paramount role in addressing micronutrient deficiencies. The promotion thereof and the augmentation of public awareness are indispensable for implementation. The advancement of big data presents challenges due to the plethora of information and the constrained processing capacity, thereby inducing difficulties in consumer decision-making. The study is obliged to intensify information dissemination to empower consumers to apprehend the value of selenium-enriched products as an integral constituent of positive nutrition guidance. The study undertook an experiment related to nutrition information acquisition, in which participants provided relevant interferences. The study utilized the structural equation model (SEM) and fuzzy set qualitative comparative analysis (*fsQCA*) to analyze the data. The study arrived at three research conclusions. Firstly, the furnishing of valuable information constitutes a significant factor in motivating consumers to purchase selenium-rich agricultural products. Secondly, the communication of brand information holds crucial significance in shaping the perception of product advantages and plays a salient role in the promotion and construction of selenium-rich agricultural products. Finally, the dissemination of health information can be incorporated into the process of promoting selenium-rich agricultural products. This conforms to the urgent necessity to address hidden hunger and establish a value identity.

## 1. Introduction

Micronutrient deficiency is prevalent in the two billion poorest individuals worldwide, impacting human health and serving as a significant contributor to *hidden hunger* [1]. In order to address this issue, the use of microorganism technology offers the potential for widely accessible and sustainable solutions [2]. This can be achieved by integrating traditional and modern biotechnology breeding methods to enhance the levels of micronutrients in food crops [3] and combat nutritional imbalance [4]. Selenium plays a crucial role in maintaining human health [5] due to its significant impact on antioxidant metabolism [6], immunity function enhancement [7], and prevention of cardiovascular diseases [8]. However, selenium is an indispensable trace element for the human body, and either insufficient or excessive selenium levels can give rise to physical discomfort [9]. Severe selenium deficiency is closely associated with the prevalence of endemic Keshan disease [10]. Introducing selenium microorganisms into food crops is an effective approach to addressing human selenium deficiency issues [11]. Furthermore, there is a necessity to enhance the promotion and public awareness of this program.

However, the emergence of the big data era has presented a significant paradox in terms of the impact of vast amounts of information resources and valuable data on consumption, especially when considering the constraints imposed by limited processing power. On the demand side of information, consumers often face an information disadvantage when it comes to the transmission of commodity information. This can hinder their decision-making process when considering the adoption of new products [12]. Compared to commonly used nutritional supplements such as iodized salt, iron soy sauce, and fortified rice, there is still a lack of awareness regarding the cognition and recognition of selenium-rich agricultural products [13]. In the absence of selenium public education, a significant number of consumers tend to hold skeptical attitudes toward the significance of selenium supplementation [14]. Therefore, it is necessary to break through the constraints of information, and there is an urgent need for positive health information guidance to facilitate consumers’ insights into the value of selenium-enriched agricultural products.

Scholars in related fields have come to recognize that information resources can serve as potential barriers to consumer attitudes and purchase intentions. The variety of channels for transmitting information facilitates the adoption and promotion of goods [15,16,17]. We cannot ignore the fact that there is a significant disparity in market information regarding selenium-rich agricultural products. Information asymmetry, misleading information, and ineffective communication all contribute to consumer resistance when making purchasing decisions. It is essential to implement targeted information interventions pertaining to selenium-rich agricultural products in order to address these obstacles and better serve consumers. The question remains whether these findings can alter the perception of the health benefits of selenium-rich agricultural products, enhance awareness, and offer guidance to influence purchasing decisions. The existing research on nutritional information interventions and the adoption of nutrition-strengthening agricultural products by consumers appears to have contradictory findings. Stadler et al. (2010) conducted randomized controlled trials to investigate the impact of information sources’ reliability on consumer behavior [18]. Meanwhile, Zhao et al. (2024) discovered that implementing nutritional information intervention significantly increased consumers’ purchasing behavior of green agricultural products [19]. Other studies have demonstrated that the provision of acceptable and valuable nutritional information is a crucial factor in the promotion of nutrition-strengthening products [20,21]. A comprehensive understanding of the potential impact of nutritional information interventions on increasing consumer awareness and acceptance of selenium-rich agricultural products, particularly when these interventions offer valuable information and complete disclosure, remains to be fully determined in academic research.

Domestic and international scholars have conducted comprehensive investigations into the impact of information interventions on consumer adoption, particularly in the context of agricultural brand development. However, there is still a lack of in-depth research on consumer food safety information for nutrition-enhancing products, especially selenium-rich agricultural products. Therefore, in order to achieve the dual objective of micronutrient supplementation and the development of a selenium-enriched agricultural industry, this paper aims to conduct research on brand building for selenium-enriched agricultural products within the theoretical framework of *Nutrition Information Intervention–Food Safety Perception–Consumer Adoption*. The focus is on constructing a theoretical model to explore the most effective ways of providing nutritional and safety information based on the core feature of selenium-enriched agricultural products—their health attributes. Specifically, we aim to examine the positive impact mechanism between health attributes and food safety on brand building for selenium-enriched agricultural products.

This paper consists of six parts.
In Section 1, we provide a background and purpose for our research, as well as propose strategies to enhance consumer information about selenium-rich foods and increase awareness of selenium-rich agricultural products. Additionally, we identify scientific challenges related to this topic.In Section 2, we construct the theoretical model, namely *Nutrition Information Intervention–Food Safety Perception–Consumer Adoption*, and propose the research hypothesis.In Section 3, we present an experimental method involving the manipulation of access to nutritional information. Participants are asked to answer questions regarding various means of obtaining information on selenium-rich agricultural products available for purchase.In Section 4, we will discuss the research hypotheses, the data analysis, and the results of the experimental tests.In Section 5, a fuzzy qualitative comparative analysis is conducted to explore the consumer’s willingness to purchase selenium-rich agricultural products. This analysis involves examining a variety of conditions and combined variables for structural analysis.The final section includes the presentation of experimental results and the analysis of the *fsQCA* (Fuzzy-set Qualitative Comparative Analysis, which is a new analytic technique that uses Boolean algebra to implement principles of comparison used by scholars engaged in the qualitative study of macro social phenomena) results, as well as offering policy recommendations.

Our goal is to enrich the theoretical framework, uncover underlying mechanisms, and provide policy recommendations that fill gaps in understanding consumer behavioral choices regarding functional agricultural products in today’s digital economy era characterized by information intervention.

## 2. Theoretical Review and Research Hypothesis

### 2.1. Impact of Information Interventions on Consumers’ Perceptions of Safety of Selenium-Rich Agricultural Products

Information interventions are intended to modify specific behavior [22], typically at a lower cost [23] and with improved outcomes [24]. In the market for selenium-rich agricultural products, there exists information asymmetry among consumers [25], as data on selenium-rich, iron-rich, and other nutrient-enhanced agricultural products is primarily confined to academic studies. Additionally, there is a lack of highly influential brand information available [26].

Consumers often encounter challenges in obtaining real-time, effective information [27] regarding nutrient-enhanced agricultural products [28], which ultimately places them at a disadvantage in the market [29]. As a result, it is important to implement interventions that encompass emotional, sensory, behavioral, and knowledge-based information in order to encourage the consumption of selenium-rich agricultural products. These products provide antioxidant properties, are resistant to arsenic and cadmium toxicity, aid in preventing a decline in thyroid function and postpartum thyroiditis, and offer valuable consumer guidance. It is essential for information intervention to have a positive effect. When consumers have access to nutritional value and safety information about selenium-rich agricultural products, their perceptions of these products are likely to be favorable. Therefore, we posit the following hypothesis:

**H1:** *nutritional information interventions on selenium-rich agricultural products for the consumer have a significant positive influence on food safety awareness*.

Emotional interventions enable consumers to engage in the production process of agricultural products [30], which serves to evoke feelings, and stimulate consumer resonance [31], which serve to evoke feelings, stimulate consumer resonance [32], and create an emotional recognition of the perceived value of the product and safety of the product [33]. Presenting participants with a video showcasing selenium-rich agricultural products and providing universal information related to food nutrition values has been shown to influence participant attitudes. Positive perceptions of the nutritional value and safety of selenium-rich agricultural products are likely to result in more favorable consumer attitudes toward these products. Based on this, we hypothesize the following:

**H1a:** *emotional interventions about selenium-rich agricultural products have a significant positive influence on food safety awareness*.

Sensory intervention provides consumers with a sensory experience that encompasses visual, auditory, olfactory, and tactile product interactions [34], thereby eliciting an aesthetic response and stimulating positive perceptions of the product’s value [35,36]. The majority of consumers are not experts, and they assess the food microbiological standards of products based on color and appearance, which is wrong, given food safety and quality. The sensory experience of selenium-rich agricultural products can be enhanced through promotional materials, graphic displays, unique fragrances, imagery, and descriptive language to convey the value of these products in a multi-sensory manner [37]. Positive information about the nutritional value and safety of selenium-rich agricultural products is anticipated to improve consumer perception, leading to the following hypothesis:

**H1b:** *sensory interventions about selenium-rich agricultural products have a significant positive influence on food safety awareness*.

Behavioral interventions involve encouraging consumers to engage in direct product experiences [38], which then translate into perceptible everyday life experiences [39] and interactions with other consumers [40]. These behavioral experiences have a lasting impact and are difficult to forget [41], and positive experiences can lead consumers to develop a value identity that motivates changes in behavior for the better [42]. The effectiveness of behavioral interventions directly influences consumers’ perceptions of selenium-rich agricultural products, as well as their overall positive experience regarding nutritional value and safety. Therefore, we hypothesize the following:

**H1c:** *behavior interventions about selenium-rich agricultural products have a significant positive influence on food safety awareness*.

Knowledge intervention refers to the active engagement of consumers [43] in the pursuit of creative thinking and practical problem-solving experiences [44]. Access to comprehensive knowledge about selenium-rich health benefits can promote curiosity and empathy [45], thereby motivating consumers to enhance their understanding of the value associated with selenium-rich agricultural products. By increasing consumer knowledge, it is possible to improve perceptions regarding the nutritional value and safety aspects of selenium-rich agricultural products. Therefore, we hypothesize the following:

**H1d:** *knowledge interventions about selenium-rich agricultural products have a significant positive influence on food safety awareness*.

### 2.2. Information Intervention and Purchase Intention

By manipulating the level of nutritional information acquisition for different groups, participants were instructed to answer questions regarding the impact of selenium-rich agricultural products on their purchase decisions under four different types of interventions: emotional, sensory, behavioral, and knowledge interventions. Purchase decisions are primarily influenced by purchase intention [46], while consumers’ willingness to purchase is highly subjective and focuses on experience in order to assess the perceived benefits and risks of the product [47]. Positive nutritional information plays a crucial role in disseminating new food technologies [48], as it can help counteract incorrect decision-making caused by delays in information acquisition resulting from information asymmetry [49], and also challenge existing perceptions to ultimately alter consumer decisions. Therefore, we hypothesize the following:

**H2:** *information intervention positively affects purchase intentions*.

**H2a:** *emotional intervention positively affects purchase intentions*.

**H2b:** *sensory intervention positively affects purchase intentions*.

**H2c:** *behavioral intervention positively affects purchase intentions*.

**H2d:** *knowledge intervention positively affects purchase intentions*.

### 2.3. Information Intervention and Value Identity

The concept of value identity pertains to the connection between an individual and an object or idea during the process of stimulation, resulting in resonance and a deep emotional experience [50]. When individuals are in a state of flow experience, they become fully engaged in their current situation [51] and user experience [52]. The utilization of publicity graphics and materials designed to engage the senses of sight, hearing, touch, and smell in promoting selenium-rich agricultural products leads consumers to perceive and acknowledge their usefulness and necessity. Therefore, we propose the following hypothesis:

**H3:** *information intervention affects value identity*.

**H3a:** *emotional intervention affects value identity*.

**H3b:** *sensory intervention affects value identity*.

**H3c:** *behavioral intervention affects value identity*.

**H3d:** *knowledge intervention affects value identity*.

### 2.4. Value Identity and Purchase Intention

Studies have indicated that enhancing consumer value identity has a positive impact on their purchase intentions [53]. When promoting selenium-rich agricultural products, the dissemination of the concept of nutrition and health can effectively address the urgent need to alleviate hidden hunger and foster a sense of value identity [54]. Consumers’ decisions to purchase selenium-rich agricultural products are influenced by various levels of identity, which consequently shape their motivation and behavior [55]. A stronger value identity is associated with a greater drive to form purchase intentions. Therefore, the following hypothesis is put forward:

**H4:** *value identity affects purchase intention*.

### 2.5. Intermediary Role of Value Identity

By disseminating nutritional information on selenium-rich agricultural products through various intervention methods, the attention of consumers can be attracted to promote these products [56]. When consumers recognize the importance of selenium-rich agricultural products in maintaining human health, they will have a more immersive experience [57], thus enhancing their awareness of these products and ultimately influencing their purchasing decisions [58]. Therefore, we hypothesize the following:

**H5:** *value identity plays an intermediary role in the relationship between information intervention and intention to purchase selenium-rich agricultural products*.

**H5a:** *value identity plays an intermediary role in the relationship between emotional intervention and intention to purchase selenium-rich agricultural products*.

**H5b:** *value identity plays an intermediary role in the relationship between sensory intervention and intention to purchase selenium-rich agricultural products*.

**H5c:** *value identity plays an intermediary role in the relationship between behavioral intervention and intention to purchase selenium-rich agricultural products*.

**H5d:** *value identity plays an intermediary role in the relationship between knowledge intervention and intention to purchase selenium-rich agricultural products*.

### 2.6. Regulatory Effect of Information Intervention Behavior

Information intervention plays a crucial role in shaping consumer value identity and serves as a decisive factor for consumers when choosing to purchase selenium-rich agricultural products. This paper argues that, compared to those individuals who did not receive any information intervention, participants who were exposed to nutritional information interventions exhibited a heightened awareness of the nutritional benefits associated with selenium-rich agricultural products. As a result, they demonstrated greater levels of value identification and showed an increased intention to make purchases. Therefore, we propose the following hypothesis:

**H6:** *the influence of value identity on the willingness to buy selenium-rich agricultural products is regulated by the effect of information intervention behavior*.

In the context of information asymmetry, this paper constructs a research model (Figure 1) that examines the impact of different nutrient information interventions on consumers’ willingness to purchase selenium-rich agricultural products.

## 3. Experimental Design and Data Analysis

### 3.1. Experimental Design

#### 3.1.1. Experimental Group and Participants

The study established an experimental group and a control group for the study. The experimental group received nutritional information on selenium-rich products, while the control group did not receive any intervention. The dependent variable was the intention to purchase selenium-rich agricultural products.

Most of the research on consumer purchase decisions utilized the experimental method, with the subjects often being college or graduate students [59,60]. The use of student samples as subjects was not only convenient but also ensured good internal validity for establishing causal effects [61]. The experimental method placed greater emphasis on randomized controlled trials (*RCTs*), where changes in the independent variable were causally linked to differentiation, rather than focusing on the issue of sample representativeness [62]. In terms of consumer awareness and recognition of selenium-rich agricultural products, it was essential to link sales to the nutritional value of the products in order to address urgent needs and resolve key points and difficulties. This paper focused on two main considerations for subject selection. On one hand, the general public was not experts in consumption and had limited access to effective information about the biological mechanisms of selenium-enriched agricultural products. Consequently, they faced difficulties in obtaining relevant information. Additionally, consumers had difficulty accessing relevant operational information regarding selenium-rich products.

Therefore, the selection of a high level of cultural and natural background was of significant importance for both undergraduate and graduate students in colleges and universities. Additionally, it was noteworthy that students constitute a significant consumer group targeted for selenium-rich products. Our research has revealed that individuals with higher levels of education exhibit greater concerns regarding food safety and overall health [63]. As such, it could be anticipated that future developments in agriculture will be heavily influenced by today’s student population.

We recruited a total of 300 undergraduate and graduate students to participate in our experiments, from whom we collected 269 valid samples. There were 130 participants in the experimental group and 139 in the control group. The aim of our study was to examine the differences in value perceptions of selenium-rich agricultural products between the intervention group and the control group, as well as to investigate whether individuals could recognize and adopt these values. Each participant was only allowed to take part in one test in order to ensure the effectiveness and integrity of the experiment. All study participants provided informed consent, and the study design was approved by the appropriate ethics review board" in this section.

#### 3.1.2. Experimental Materials Design


Questionnaire design. The questionnaire was designed separately to measure the impact of an information intervention on consumers’ willingness to purchase and how it affected their behavior. This involved creating an experimental manual for the information intervention.Correction and modification of the questionnaire.Questionnaire testing. We randomly selected two classes of students to pilot the questionnaire in order to further enhance its quality.Appropriate subjects were selected in accordance with the sampling method for the inter-group experiment.


#### 3.1.3. Experimental Process


Before the test. In order to assess validity and ensure homogeneity among the participants, pre-cognitive testing was utilized to measure subjective knowledge of selenium-rich agricultural products. This was conducted to ensure that there were no significant differences across the sample. The knowledge test questionnaire was based on previous studies [64] and used a 5-point Likert scale. The questions included *Do you know anything about selenium?*, *Do you think people need selenium?*, and *What do you know about selenium-rich agricultural products?* The results indicated that there were no significant differences between the experimental and control groups (*M*_experiment group_ = 2.52, *M*_control group_ = 2.41, *t* = 1.633, *p* > 0.050).Prescribed experiments. Firstly, two separate classrooms were designated for the experimental and control groups. The experimental group watched promotional videos, viewed publicity graphics, read relevant materials, and listened to the host provide details outlining the nature of selenium-rich agricultural products and their advantages. Subsequently, a questionnaire was distributed to the experimental group without allowing any discussion during its completion. After 10 min, the questionnaires were collected. The control group also completed the questionnaire without discussion or input from the host and had 10 min to submit their responses.


### 3.2. Scale Design

The questionnaire utilized for data collection included demographic information such as gender, age, education background, rural life experience, and seven latent variables adapted from the relevant literature. The variables primarily utilized a Likert 5-point scale. The relevant measures and reference sources are shown in Table 1.

### 3.3. Descriptive Statistics

The test sample consisted of 269 individuals, with 110 males (40.9%) and 159 females (59.1%). The majority of the participants (86.2%) were in the 19–35 age group, which was expected for a sample of students. In the experimental group (*N* = 130), there were 63 men (48.46%) and 67 women (51.54%), while in the control group (*N* = 139), there were 47 men (33.81%) and 92 women (66.19%). A high percentage of participants in both groups had experience of rural life, with it being 77.69% in the experimental group and 74.10% in the control group. In terms of monthly income, 76.15% of the experimental group earned 2000 yuan or less, while in the control group, it was 83.45%, indicating only a slight difference between the groups. Regarding monthly expenditure, 76.15% of the experimental group spent 2000 yuan or less, compared to 69.78% in the control group, once again demonstrating a small difference between the groups. With regards to purchasing selenium-rich products, 74.62% of the experimental group did not buy selenium-rich agricultural products, as opposed to 74.82% in the control group. The homogeneity of the two groups was strong (Table 2).

### 3.4. Inspection and Reliability and Validity of Exploratory Factor Analysis

Smart PLS 2.0 was utilized to examine the data of seven variables, including subjective knowledge and emotional, sensory, behavioral, and knowledge intervention. The results are shown in Table 3. The Cronbach’s *α* values ranged from 0.602 to 0.905, all of which exceeded the threshold of 0.600, indicating good reliability of the scale. Furthermore, a reliability test was conducted after removing each variable measure individually, resulting in only minor changes to the Cronbach’s *α* values, thus confirming that the scale’s reliability was acceptable. The standard variable load exceeded 0.600, with *AVE* (Average Variance Extracted) values ranging from 0.553 to 0.799, indicating a favorable level of convergent validity. The minimum value was 0.787, and the *CR* (Composite Reliability) value exceeded the critical threshold of 0.700, confirming the internal consistency of the questionnaire.

The distinction validity test is shown in Table 4, indicating that the correlation coefficients of the variables were smaller than the square root of the average extracted variance for each factor. This suggested that there was good validity in distinguishing between variables.

The exploratory factor analysis indicated that *KMO* (Kaiser–Meyer–Olkin-Measure of Sampling Adequacy; this test was employed to contrast the relative magnitudes of the simple and partial correlation coefficients among the original variables) value was 0.925, and the *p*-value of Bartlett’s sphericity test was 0.000, suggesting that factor analysis is highly feasible for the data.

## 4. Hypothesis Testing and Results of Analysis

### 4.1. Information Intervention and Consumer Food Safety Perception Hypothesis Test

Based on the results of the independent sample *t*-test results shown in Table 5, there were significant differences in consumer perception of food safety and nutritional value with emotional, sensory, behavioral, or knowledge interventions. The mean values indicated that groups receiving nutrition information intervention had a stronger perception of the safety of selenium-rich agricultural products compared to those without the intervention. Therefore, H1 was supported.

### 4.2. Main Effect Inspection

When examining the direct-action mechanism of information intervention and purchase intention, as well as the impact of value identity on purchase intention, hypotheses were tested using SPSS. The data was sampled 5000 times with bootstrapping to obtain correlation path coefficient and significance level. The results are shown in Table 6.

Emotional, sensory, behavioral, and knowledge interventions demonstrated significant positive effects on consumers’ intention to purchase selenium-rich agricultural products. Specifically, emotional intervention had a positive influence on purchase intention (*β* = 0.29, *p* < 0.001), supporting hypothesis H2a. Sensory intervention also had a positive influence on purchase intention (*β* = 0.251, *p* < 0.001), validating hypothesis H2b. The behavioral intervention positively impacted purchase intention as well (*β* = 0.279, *p* < 0.001), confirming hypothesis H2c. Finally, knowledge intervention was found to positively influence purchase intention (*β* = 0.285, *p* < 0.001), thus supporting the validity of hypothesis H2d.

Emotional, sensory, behavioral, and knowledge interventions all exerted a significant, positive influence on consumer value identity. Emotional intervention (*EI*) influenced the value of identity (*β* = 0.426, *p* < 0.001), supporting H3a. Sensory intervention (*SI*) influenced the value of identity (*β* = 0.463, *p* < 0.001), supporting H3b. Behavioral intervention (*BI*) influenced the value of identity (*β* = 0.554, *p* < 0.001), supporting H3c. Knowledge intervention (*KI*) influenced the value of identity (*β* = 0.725, *p* < 0.001), assuming support for H3d holds true. The effect of value identity (*VI*) on purchase intention (*PI*) of selenium-rich agricultural products was significantly positive (*β* = 0.586, *p* < 0.001), assuming support for H4.

### 4.3. Mediation Effect of Inspection

SPSS (Statistical Product and Service Solutions. The IBM^®^ SPSS^®^ Statistics V28.0.0 software platform offers advanced statistical analysis, a vast library of machine learning algorithms, text analysis, open-source extensibility, integration with big data, and seamless deployment into applications) was employed to conduct a stepwise regression analysis in order to examine the potential mediating role of value identification between the four facets of information intervention and purchase intention. The findings are presented in Table 7. As indicated by steps 1 and 2, and subsequently by the dependent variable in step 3 (i.e., purchase intention), along with the independent variables (emotional intervention, sensory intervention, behavioral intervention, knowledge intervention), the regression analysis of the mediating variable (value identity) revealed that the β value for emotional intervention was 0.113, corresponding to a *β* value for identity of 0.315, with *p* < 0.010. Sensory intervention has a *β* value of 0.121, with a corresponding recognition *β* value of 0.289, and *p* < 0.001. Behavioral interventions exhibited a *β* value of 0.084, coupled with an identity *β* value of 0.310 at *p* < 0.010. Knowledge intervention yielded a *β* value of 0.142 coupled with an identity *β* value if 0.263 at *p* < 0.001. Upon incorporating mediator variables into consideration, there was a reduction in regression coefficients for independent variables, but values of *β*_3_ and *β*₄ remain highly significant, thus indicating that identified values pertaining to information intervention had a moderate partial mediation effect on purchase intention, thereby lending support to hypotheses H5a, H5b, H5c, and H5d.

According to Table 8, the total effects of emotional, sensory, behavioral, and knowledge interventions on purchase intention were 0.290, 0.251, 0.279, and 0.285, respectively. Upon introducing the intermediary variable value identification, the direct effect of purchase intention on the dependent variable decreased. This suggested that value identity partially intervened in the information intervention of the four factors for purchasing selenium-rich agricultural products. The substantial effect proportion indicated that a more obvious role of information intervention made it easier to stimulate emotional resonance among consumers and produce value identification, thereby promoting consumer decision-making in favor of purchasing selenium-rich agricultural products.

### 4.4. Regulating Effect Inspection

To examine the regulatory impact of informative intervention behavior on the relationship between subject value identity and purchase intention among experimental and control groups, firstly, we demonstrated the effectiveness of the information intervention through independent sample t-test results, showing a significant difference in value identity between the experimental group and control group, with higher group value identity in the experimental group (*M*_Experimental Group Value identity_ = 3.99, *M*_Control identity_ = 3.21, *p* < 0.001). Secondly, SPSS statistical analysis software was utilized to standardize relevant variables and obtain original variables through interaction. Finally, stratified regression analysis was conducted to predict regulatory effects, as indicated in Table 9.

Model 1 only included gender, age, rural experience, occupation, income, consumer spending, and subjective knowledge as control variables. The idiosyncratic knowledge of selenium-rich agricultural products professionals and consumers to purchase intention showed a significant influence. Model 2’s control variables, independent variables, and adjustments—professional, VI, and KI—had a significant impact on purchase intention. Model 3’s control variables, independent variables, adjustment, and interaction showed that the *x*-value VI interaction interventions had a significant influence on PI (*t* = 4.255, *p* < 0.001), indicating that hypothesis H6 was supported.

## 5. Qualitative Comparison Analysis and Fuzzy Sets

In the structural equation model based on empirical evidence, this paper employed *fsQCA* (fuzzy qualitative comparative analysis) methods to evaluate the purchasing intentions of selenium-rich agricultural products. Through its integration of quantitative and qualitative characteristics, *fsQCA* examined the influence of multiple variables on outcome variables from a configurational perspective, thus enriching and deepening research conclusions.

### 5.1. Selection and Calibration of Variables

Based on the aforementioned empirical studies, this paper has chosen emotional intervention, sensory intervention, behavioral intervention, knowledge intervention, value identity, and the subjective knowledge of consumers as the independent variables. The outcome variable was the intention to purchase selenium-rich agricultural products which was analyzed using *fsQCA* 3.0 software for fuzzy set qualitative comparative analysis.

In order to improve the interpretability of the results, it was important to calibrate the variable data before analysis. The average value of the six antecedent variables was evaluated in Excel. Following Ragin’s calibration method (2009), the terciles quantiles of 5%, 50%, and 95% were utilized as thresholds for completely non-membership, crossover, and complete membership in calibrating the relevant variable data [84]. Subsequent to data calibration, *fsQCA* software was employed to analyze the necessity and adequacy of each individual antecedent variable. As shown in Table 10, none of the pre-dependent variables exhibited a consistency greater than or equal to 0.8 for the outcome variable. This preliminary assessment indicated that the interpretative power of individual variables on the results was weak, suggesting that there were no necessary or sufficient conditions influencing consumers’ decision to purchase selenium-rich agricultural products. Consequently, a combination of variables required various conditions for structural analysis, and further investigation into factors affecting consumers’ decision-making regarding selenium-rich agricultural products would be essential in identifying this combination.

### 5.2. Analysis of fsQCA Results

In the antecedent and consequence variables after assignment, as the basis for analyzing *QCA*, a truth table should be constructed. The consistency threshold was set at 0.8, the case frequency threshold was set at 1, and the *PRI* (Proportional Reduction in Inconsistency; it functions as an alternative metric for consistency in subset relationships) threshold was set at 0.75. Based on the truth table above, *fsQCA* 4.1 software (Copyright ^©^, Prof. Charles C. Ragin, University of California, Irvine, USA) typically generates a simple, intermediate, and complex solution after standardization. The combination of simple and intermediate solutions constituted purchasing selenium-rich agricultural products before configuring the dependent variable’s will. The results are presented in Table 11.

When subjected to information intervention, a total of six configurations were examined. The overall consistency achieved was 0.859803, with an overall coverage rate of 0.554172, indicating a relatively effective interpretation effect of the six configurations. The specific details of the six configurations are presented below.
Emotional, sensory, and behavioral interventions could influence consumers to purchase selenium-rich agricultural products, even without subjective knowledge of the products.When consumers had subjective knowledge of selenium-rich agricultural products, sensory, behavioral, and knowledge interventions stimulated their desire to buy the products.The emotional intervention stimulated vitality in consumers and influenced their recognition of value when purchasing selenium-rich agricultural products through emotional and sensory intervention.Consumers were motivated to buy selenium-rich agricultural products through sensory and knowledge intervention, as well as value identification under the influence of sensory, behavioral, and knowledge interventions.Even without any behavioral intervention, consumers with subjective knowledge of selenium-rich agricultural products may still be stimulated to purchase through emotional and knowledge interventions.Subjective knowledge alone could motivate consumers to buy selenium-rich agricultural products regardless of emotional or sensory intervention.

## 6. Discussion and Conclusion

### 6.1. Further Discussions

Three specific points merit further discussion.
In the context of asymmetric information, providing valuable information on nutrition interventions related to selenium-rich agricultural products to promote food safety awareness may incentivize consumers to make purchases. Emotional interventions that highlight the nutritional value of selenium-rich agricultural products have the most direct impact on consumer purchase decisions [19]. In essence, as a dynamic communication tool, promotional videos not only provide a comprehensive and intuitive presentation of selenium-rich agricultural products but also evoke strong emotional responses. The conclusion further confirms hypotheses 1–4. Emotions and values are more relatable and can drive consumer emotions while enhancing their desire to make a purchase. The next most significant impacts on purchase intention come from knowledge and behavioral interventions, indicating that although informative text is closely associated with sensory stimuli such as vision, smell, and taste, these interventions are no longer as effective due to an overload of information [23]. Therefore, in promoting selenium-rich agricultural products, marketers should prioritize emotional intervention followed by knowledge intervention, then behavioral intervention, and, finally, sensory intervention.Value recognition plays a crucial role in influencing the purchase decision of selenium-rich agricultural products and acts as a partial mediator in the relationship between the four factors of information intervention and purchase intention [34]. The conclusion further confirms hypothesis 5. This indicates that nutritional information interventions can effectively drive consumers towards purchasing these products.Information intervention behavior positively influences the causal relationship between value identity and purchase intention, suggesting that the experimental group with interventions performs better than the group without them. This conclusion further confirms the theoretical model. Information intervention significantly enhances value identity, which, in turn, strengthens purchase intentions.

### 6.2. Theoretical Contributions

Valuable information plays a crucial role in shaping consumers’ perceptions of agricultural products rich in selenium, thus influencing their purchase decisions. On the contrary, invalid or asymmetrical information does not aid consumers in making rational choices. Therefore, complete, truthful, and effective product information directly impacts consumer perceptions.

Furthermore, brand communication significantly influences consumers’ perception of product advantages and contributes to the branding and promotion of selenium-rich agricultural products. The health benefits associated with these products are a core advantage that can be emphasized through valuable nutritional information. This includes highlighting their role in antioxidant metabolism, immune system enhancement, and cardiovascular disease prevention, thus illustrating the practicality and necessity of selenium-rich agricultural products.

Moreover, within the realm of brand promotion, the dissemination of health information aligns with the pressing need to address “hidden hunger” and establish a valued identity. Providing positive guidance through nutritional information can heighten consumers’ emotional resonance, sensory experience, community influence, and rational thinking in order to establish an emotional connection with the product. By utilizing informational (such as test reports for selenium-rich agricultural products, certification reports for selenium-rich agricultural products, etc.) and emotional (such as promotional videos for selenium-rich agricultural products, positive consumer reviews, etc.) interactions, we can enhance the perceived value of selenium-rich agricultural products, mobilize consumer health expectations, gain recognition from consumers, and ultimately stimulate their purchasing intention.

### 6.3. Policy Suggestions

Selenium-rich agricultural products, as a type of nutrition-strengthening food, are characterized by a certain level of professionalism. If consumers do not have access to accurate product information, they may be reluctant to make a purchase and could even develop negative attitudes toward the products. Consequently, it is essential to address issues of information asymmetry in order to enhance consumer adoption of selenium-rich agricultural products and improve associated brand construction. To this end, the following specific policy suggestions are provided:The government should take the initiative to establish a reliable, comprehensive, and transparent information service platform for selenium-rich agricultural products to combat misinformation. Meanwhile, enterprises should utilize big data, artificial intelligence, and other digital technologies to effectively gather and integrate valuable information related to selenium-rich agricultural products for consumers. Meanwhile, the government is obligated to establish a traceable system covering the entire supply chain and allocate codes to enterprises based on the cloud. Consumers have the ability to scan these codes to acquire information regarding the entire supply chain production process, thereby achieving comprehensive transparency in both production and supervision. Consumers themselves should carefully consider the information they receive as their scientific understanding of selenium-rich agricultural products improves, using it as a reference for their purchasing decisions.Enterprises offering selenium-rich products should clarify the product advantages and enhance the communication and appeal of product information. They should expand communication channels to include television, newspapers, and radio, as well as new media platforms such as WeChat TM, TikTok TM, Weibo TM, and Facebook TM to disseminate valuable information regarding selenium-rich agricultural products. Furthermore, they should leverage the influence of experts and community leaders in spreading this information.In terms of exploring emotional connections between brands and consumers, selenium-rich enterprises should guide consumer understanding of selenium-rich agricultural products. They ought to adjust publicity methods based on customer feedback while promoting an appreciation for the value of the products. Enterprises must also provide consumers with diverse and personalized channels for receiving information, thereby strengthening emotional resonance with consumers and enhancing alignment between customers’ preferences and products.

### 6.4. Conclusions

Based on the principles of information symmetry and consumer acceptance, this article primarily employed an experimental method to investigate valuable insights into the impact of nutrition interventions on consumers’ purchase of selenium-rich agricultural products. It also delved into the concept of value identity, exploring how mechanisms and information influenced consumers’ perception of value. Furthermore, the article analyzed the factors that trigger consumers to purchase selenium-rich agricultural products using *fsQCA*, focusing on different configurations. A comprehensive analysis follows.

First, the SEM analysis indicated that implementing various interventions to provide consumers with valuable nutritional information about selenium-rich agricultural products effectively enhances their perception of product safety. This approach also led to improvements in knowledge and recognition, thereby encouraging them to purchase selenium-rich agricultural products.

Secondly, based on the analysis of *fsQCA*, we discovered that stimulating consumers to purchase selenium-rich agricultural products reveals two patterns (refer to Table 11). Mode 1 indicated that when there were three information intervention methods at a good level (configurations A1-A4), consumers would be motivated to buy selenium-rich agricultural products regardless of their access to subjective knowledge and value identification. On the other hand, Mode 2 (configurations A5-A6) suggested that when consumers have subjective knowledge of the products, only interventions focusing on knowledge would trigger their willingness to make a purchase. Comparing the two models, it could be noted that Model 1 has a higher coverage rate than Model 2 and also has greater explanatory power.

## Figures and Tables

**Figure 1 foods-13-03256-f001:**
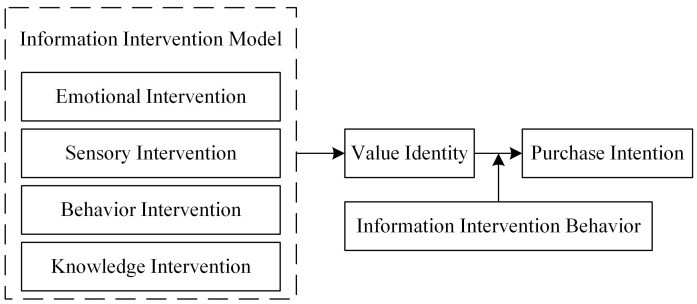
Research model.

**Table 1 foods-13-03256-t001:** Experimental measurements and their sources.

Variables	Measure Item	References
Subjective knowledge (*SK*)	*SK*_1_: Do you know anything about selenium?	Giller, 2021 [65]
	*SK*_2_: Do you think people need selenium?	Wang, 2022 [66]
	*SK*_3_: How much do you know about selenium-rich agricultural products?	Liu et al., 2019 [64]
Emotional intervention (*EI*)	*EI*_1_: After watching the video, it is clear that selenium deficiency will cause certain diseases.	Radomska et al., 2021 [67] Winther et al., 2020 [68]
	*EI*_2_: The video says that the five longevity zones are soil-based selenium-rich areas.	
	*EI*_3_: Selenium is an essential trace element for the human body.	
Sensory intervention (*SI*)	*SI*_1_: In the figure, selenium-rich navel oranges feel good and are sweet.	Gómez et al., 2021 [69]
	*SI*_2_: Selenium-rich rice has crystal clear particles and a mellow taste.	Le et al., 2020 [30]
	*SI*_3_: In the figure, selenium-rich sweet potatoes have thin skins, red meat, and a rich aroma.	Pyo et al., 2022 [70]
Behavioral intervention (*BI*)	*BI*_1_: Selenium-rich blueberries are aesthetically pleasing and boost immunity. Selenium-rich experts to promote propaganda.	Lin et al., 2021 [71]
	*BI*_2_: Selenium-rich blueberries are aesthetically pleasing and boost immunity. Star big V promotion propaganda.	Hong et al., 2020 [72]
	*BI*_3_: In the figure, customers in the store snapped up selenium-rich agricultural products.	Takahashi et al., 2020 [73]
	*BI*_4_: Friends and relatives have bought selenium-rich agricultural products.	Xiao et al., 2022 [53]
	*BI*_5_: Selenium-rich agricultural products are starting to become popular.	Ren et al., 2021 [74]
Knowledge intervention material	What is selenium? (1) Although the Chinese people are full, 300 million people lack trace elements, which can lead to malnutrition, and they lose a lot of money every year. (2) Scientists say selenium can fight cancer, is good for the eyes and the cardiovascular system, prevents liver disease, treats diabetes, detoxifies, has anti-aging properties, and improves fertility. (3) General Secretary Xi said that Jiangxi province should develop a selenium-rich industry, an idea strongly supported by the provincial government. Jiangxi Agricultural University established the Jiangxi Selenium-rich Agricultural Research Institute. (4) People who eat selenium-rich agricultural products also feel good about cancer-free villages and longevity villages.	
Knowledge intervention (*KI*)	*KI*_1_: Read the above materials; do you think selenium-rich foods are useful?	Wang et al., 2020 [75] Tangri et al., 2021 [76] Mahorter et al., 2020 [77]
	*KI*_2_: Read the above materials; do you agree that selenium prevents cancer?	
	*KI*_3_: Read the above materials; do you agree that selenium benefits the eyes?	
	*KI*_4_: Read the above materials; do you agree that selenium has anti-aging properties?	
Value identity (*VI*)	*VI*_1_: I think the nutritional function of selenium-rich products is credible.	Hati et al., 2021 [78]
	*VI*_2_: Se deficiency causes malnutrition; therefore, I support selenium-rich products.	Campbell et al., 2022 [79]
	*VI*_3_: I think overall, selenium-rich produce is good for your health.	Feng et al., 2020 [80]
Purchase intention (*PI*)	*PI*_1_: After participating in the experiment, I will learn more about selenium-rich agricultural products.	Lutzke et al., 2021 [81]
	*PI*_2_: After participating in the experiment, I will recommend selenium-rich agricultural products to others.	Katt et al., 2020 [82]
	*PI*_3_: After participating in the experiment, I am willing to buy selenium-rich agricultural products.	Yang et al., 2020 [83]

Note: With the exception of the selenium-rich intervention, the information provided to the control group was essentially identical to that given to the experimental group. However, it is worth noting that there was no dissemination of selenium-rich publicity videos, textual materials, or explanations by the host.

**Table 2 foods-13-03256-t002:** Sample description.

		Overall Sample		Experimental Group		Control Group	
		No.	(%)	No.	(%)	No.	(%)
Gender	Male	110	40.90	63	48.46	47	33.81
	Female	159	59.10	67	51.54	92	66.19
Age	Under the age of 18	35	13.00	8	6.15	27	19.42
	19~35 Years old	232	86.20	120	92.31	112	80.58
	36~45 Years old	2	0.70	2	1.54	0	0.00
Education	College or undergraduate	212	78.80	80	61.54	132	94.96
	Master’s degree or above	57	21.20	50	38.46	7	5.04
Have you ever had farming experience?	Yes	204	75.80	101	77.69	103	74.10
	No	65	24.20	29	22.31	36	25.90
Monthly income	RMB 2000 or less	215	79.90	99	76.15	116	83.45
	RMB 2000–4000	40	14.90	19	14.62	21	15.11
	RMB 4000–6000	6	2.20	5	3.85	1	0.72
	above	8	3.00	7	5.38	1	0.72
Monthly expenditure	RMB 2000 or less	196	72.90	99	76.15	97	69.78
	RMB 2000–4000	49	18.20	22	16.92	27	19.42
	RMB 4000–6000	15	5.60	3	2.31	12	8.63
	above	9	3.30	6	4.62	3	2.16
Purchasing behavior	No	201	74.72	97	74.62	104	74.82
	Yes	68	25.28	33	25.38	35	25.18

**Table 3 foods-13-03256-t003:** Reliability and convergent validity analysis.

Variables	Measure Items	Cronbach’s *α*	Delete the Post-Measure Values	Factor Loading	*CR*	*AVE*
Subjective knowledge	*SK* _1_	0.602	0.576	0.782	0.787	0.553
	*SK* _2_		0.518	0.691		
	*SK* _3_		0.404	0.756		
Emotional intervention	*EI* _1_	0.687	0.651	0.713	0.826	0.614
	*EI* _2_		0.576	0.816		
	*EI* _3_		0.549	0.817		
Sensory intervention	*SI* _1_	0.764	0.795	0.786	0.864	0.680
	*SI* _2_		0.596	0.859		
	*SI* _3_		0.642	0.827		
Behavioral intervention	*BI* _1_	0.877	0.863	0.790	0.911	0.671
	*BI* _2_		0.860	0.774		
	*BI* _3_		0.855	0.807		
	*BI* _4_		0.833	0.881		
	*BI* _5_		0.844	0.840		
Knowledge intervention	*KI* _1_	0.905	0.907	0.835	0.934	0.779
	*KI* _2_		0.862	0.904		
	*KI* _3_		0.871	0.892		
	*KI* _4_		0.866	0.897		
Valueidentity	*VI* _1_	0.874	0.819	0.897	0.923	0.799
	*VI* _2_		0.848	0.876		
	*VI* _3_		0.800	0.908		
Purchase intention	*PI* _1_	0.711	0.771	0.676	0.839	0.637
	*PI* _2_		0.530	0.874		
	*PI* _3_		0.534	0.832		

**Table 4 foods-13-03256-t004:** Differential validity.

	*SK*	*VI*	*EI*	*SI*	*KI*	*BI*	*PI*
*SK*	0.894						
*VI*	0.496	0.784					
*EI*	0.519	0.422 **	0.824				
*SI*	0.264	0.174 **	0.238 **	0.744			
*KI*	0.698	0.537 **	0.555 **	0.226 **	0.883		
*BI*	0.539 *	0.434 **	0.733 **	0.247 **	0.604 **	0.819	
*PI*	0.586 **	0.437 **	0.516 **	0.298 **	0.481 **	0.467 **	0.798

Note: * means *p* < 0.05, ** means *p* < 0. 01, two-tailed test. The values on the diagonal are *AVE* square root values, and the other values are correlation coefficients between the planes.

**Table 5 foods-13-03256-t005:** Independent sample *t*-test.

Variables	Measure Items	Group (Mean Value of ± Standard Deviation)		*t*
		Experimental Group (*N* = 130)	Control Group(*N* = 139)	
Emotional intervention	*EI* _1_	4.02 ± 0.87	3.65 ± 0.74	3.718 ***
	*EI* _2_	3.95 ± 0.61	3.45 ± 0.59	6.692 ***
	*EI* _3_	4.24 ± 0.72	3.82 ± 0.78	4.554 ***
Sensory intervention	*SI* _1_	3.02 ± 0.89	2.38 ± 0.84	6.041 ***
	*SI* _2_	3.22 ± 0.83	2.65 ± 0.81	5.667 ***
	*SI* _3_	3.08 ± 0.82	2.58 ± 0.77	5.177 ***
Behavioral intervention	*BI* _1_	3.37 ± 0.82	2.78 ± 0.81	5.840 ***
	*BI* _2_	2.92 ± 0.92	2.49 ± 0.87	3.887 ***
	*BI* _3_	3.16 ± 0.72	2.90 ± 0.72	2.986 **
	*BI* _4_	3.34 ± 0.69	2.91 ± 0.69	5.066 ***
	*BI* _5_	3.27 ± 0.75	2.83 ± 0.76	4.711 ***
Knowledge intervention	*KI* _1_	3.90 ± 0.71	3.43 ± 0.65	5.619 ***
	*KI* _2_	3.64 ± 0.75	3.09 ± 0.67	6.294 ***
	*KI* _3_	3.71 ± 0.79	3.08 ± 0.64	7.140 ***
	*KI* _4_	3.69 ± 0.82	3.06 ± 0.67	6.910 ***

Note: *** denoted *p* < 0.010, ** denoted *p* < 0.050.

**Table 6 foods-13-03256-t006:** Test for the significance of the pathway coefficient.

Path Relationship	Coefficient	*t*	*p*
*EI → PI*	0.290	6.777	0.000 ***
*SI → PI*	0.251	6.776	0.000 ***
*BI → PI*	0.279	7.178	0.000 ***
*KI → PI*	0.285	8.912	0.000 ***
*EI → VI*	0.426	6.584	0.001 ***
*SI → VI*	0.463	8.697	0.000 ***
*BI → VI*	0.554	10.249	0.000 ***
*KI → VI*	0.725	14.829	0.000 ***
*VI →* *PI*	0.586	14.385	0.000 ***

Note: *** denoted *p* < 0.010.

**Table 7 foods-13-03256-t007:** Value identification of the mediation effect between the four factors of information intervention and purchase intention.

Step	Explanatory Variable	Explained Variable	*β*	Establishment Conditions
Step 1	Independent variable	Dependent variable	*β*1	*β*1 should be significant
	Emotional intervention	Purchase intention	0.290 ***	
	Sensory intervention		0.251 ***	
	Behavioral intervention		0.279 ***	
	Knowledge intervention		0.285 ***	
Step 2	Independent variable	Metavariable	*β*2	*β*2 should be significant
	Emotional intervention	Value identity	0.426 ***	
	Sensory intervention		0.463 ***	
	Behavioral intervention		0.554 ***	
	Knowledge intervention		0.725 ***	
Step 3	Explanatory variable	Explained variable	*β*3	*β*_4_ should be significant *β*_3_ has no significance, The full mediation effect holds true *β*_3_ has significance Part of the intermediary effect is established
	Emotional intervention	Purchase intention	0.113 **	
	Sensory intervention		0.121 ***	
	Behavioral intervention		0.084 **	
	Knowledge intervention		0.142 ***	
	Metavariable		*β*4	
	Value identity		0.315 ***	
			0.289 ***	
			0.310 ***	
			0.263 ***	

Note: *** denoted *p* < 0.010, ** denoted *p* < 0.050.

**Table 8 foods-13-03256-t008:** Results of mediation effect tests.

Items	Conclusion	*c*	*a* × *b*	*c*’	Effect Ratio
*X*_1_-*M*-*Y*	Part of the intermediary	0.29	0.131	0.159	45.08%
*X*_2_-*M*-*Y*	Part of the intermediary	0.251	0.142	0.108	56.79%
*X*_3_-*M*-*Y*	Part of the intermediary	0.279	0.168	0.111	60.21%
*X*_4_-*M*-*Y*	Part of the intermediary	0.285	0.143	0.142	50.26%

Note: *X*_1_—emotional intervention, *X*_2_—sensory intervention, *X*_3_—behavioral intervention, *X*_4_—knowledge intervention, *M*—value identity, *Y*—purchase intention.

**Table 9 foods-13-03256-t009:** Test of regulatory effects of informative intervention behavior.

Variable Type	Variable	Model 1	Model 2	Model 3
		*t*	*t*	*t*
Control variable	Gender	0.987	1.359	1.718
	Age	0.115	−1.664	−1.808
	Rural experience	−0.901	−0.763	−1.028
	Job	−2.042 *	−2.152 *	−2.262 *
	Income	0.545	−0.696	−0.536
	Consumer expenditure	−4.13	1.554	1.591
	Subjective knowledge	4.321 ***	2.834 **	2.239 *
Independent variable	Value identity		7.143 ***	7.939 ***
Regulated variable	Group		−3.524 ***	−3.509 ***
Regulatory effect	Value identity × Group			4.225 ***
	*R* ^2^	0.126	0.375	0.416
	Δ*R* ^2^	0.100	0.351	0.391
	*F* change	4.705	51.405	17.847
	*F* change significance	0.000 ***	0.000 ***	0.000 ***

Note: *** denoted *p* < 0.010, ** denoted *p* < 0.050, and * denoted *p* < 0.100.

**Table 10 foods-13-03256-t010:** Necessity of single-antecedent variables.

Dependent Variable	Outcome Variable (Willingness to Buy)	
	Consistency	Coverage
*EI*	0.576960	0.829049
*~* *EI*	0.423040	0.685417
*SI*	0.606343	0.841471
*~* *SI*	0.393657	0.664337
*BI*	0.647222	0.821453
*~* *BI*	0.352778	0.671660
*KI*	0.458586	0.862347
*~* *KI*	0.541414	0.692927
*SK*	0.472778	0.804458
*~* *SK*	0.527222	0.726767
*VI*	0.483010	0.824761
*~* *VI*	0.516990	0.710644

**Table 11 foods-13-03256-t011:** Pre-dependent variable configurations of purchase intention.

Variable	*A* _1_	*A* _2_	*A* _3_	*A* _4_	*A* _5_	*A* _6_
*EI*	●		•		•	⨂
*SI*	●	●	●	●		⨂
*BI*	●	•		•	⨂	⨂
*KI*		●	●	●	●	●
*SK*	⨂	•			●	●
*VI*			•	•	•	⨂
Consistency	0.842	0.867	0.875	0.884	0.829	0.806
Coverage	0.310	0.265	0.349	0.367	0.132	0.116
Unique coverage	0.049	0.010	0.003	0.007	0.002	0.009
Solution consistency	0.554172					
Solution coverage	0.859803					

Note: ● indicated that the core conditions existed; • indicated an edge condition existed; ⨂ represented a missing core condition; ⨂ represented a missing edge condition; a blank indicated that the condition could exist or does not exist; *EI* = emotional intervention; *SI* = sensory intervention; *BI* = behavioral intervention; *KI* = knowledge intervention; *SK* = subjective knowledge; *VI* = value identity.

## Data Availability

The original contributions presented in the study are included in the article, further inquiries can be directed to the corresponding authors.

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
