# Peer review of "Valuable Data “Gain” and “Loss”: The Quantitative Impact of Information Choice on Consumers’ Decision to Buy Selenium-Rich Agricultural Products"

_foods, 2024, doi:10.3390/foods13203256_

Round 1

Reviewer 1 Report

Comments and Suggestions for Authors

#REVIEW COMMENTS

The authors have made a great effort to revise the manuscript with justifiable and sufficient kinds of literature where necessary to make the paper rich and more scientific. The following observations need to be addressed for acceptance and publication recommendation.

·      Abstract: lines 19 and 21, 23 kindly avoid the use of “we” and use wording such as “the study”

·      Line 41: correct “an” to “a”

·      The study entirety centers on “selenium-rich agricultural products” and I wonder why the title isn’t incorporating “selenium-rich agricultural products” for coherence but still holding on to the wording “agricultural product” which to a great extent presents a wide scope of studies relative to the agriculture field but the study does not. The write-up first presents the lack of micronutrients as the integral cause of “hidden hunger” and further explicitly acknowledges the vital merits and demerits of selenium which is a micronutrient. Further, the paper details information on “selenium-rich agricultural products” coupled with genetic programming (biotechnology) to enhance crops with selenium. Lastly, lines 86-87 “However, there is still a lack of in-depth research on consumer food safety information for nutrition-enhancing products, especially selenium-rich agricultural products”. This statement indicates the study is centered on “selenium-rich agricultural products”. Authors should kindly revise the paper title and embed “selenium-rich agricultural products” to make the manuscript more coherent and scientific.

·      Lines 136-141: is the statement indicating that the lack of information on selenium-rich agricultural products accessible to consumers influences the effect (negative or positive) of selenium-rich agricultural products consumption since line 44 states “inadequate or excessive selenium levels can cause physical discomfort”, Also, lines 54-56 states “Compared to commonly used nutritional supplements such as iodized salt, iron soy sauce, and  fortified rice, there is still a lack of awareness regarding the cognition and recognition of selenium-rich agricultural products” and line 57 states “Consumers tend to be skeptical about the nutritional value of selenium”. These statements seem to reveal that “selenium-rich agricultural products” producers reluctantly avoid providing nutritional detail on the selenium product for consumers perusal. If so, why? Is selenium presence in a product that bad to affect product sales should buyers identify it as an ingredient in a product? Again, is the skeptical nature of consumers about selenium-rich agricultural products the reason for the study? Authors should justify these sections for coherence and avoid any discrepancies among readers after publication.

·      Lines 161-162: “The majority of consumers are not experts, and they assess the food microbiological standards of products based on color and appearance.” This statement should be revised as “The majority of consumers are not experts, and they assess the food microbiological standards of products based on color and appearance which is wrong given food safety and quality”. This is because the color and appearance of food from a food science perspective don’t merit a food wholesome or microbiologically safe for consumption. Therefore, even if buyers or consumers consider these indicators of food to be “microbially safe” it could be pardoned based on their lack of expertise but to a great extent technically and scientifically it must be a “wrong” approach to judge a “safe and quality food”. Hence, the recommended revised statement could be considered and changes affected at various sections where necessary.

·      Line 209: the wording “state of flow” seems confusing. Kindly revise to a better word for clarity.

·      Lines 237-246: instead of H5, H5a, H5b, H5c, and H5d why not consider H5a, H5b, H5c, H5d and H5e

·      Line 267: change “we” and “our” to “the study”

·      Line 342: there must be a space after the period in “(66.19%).A” to “(66.19%). A”

·      Line 351: delete one of the periods “(Table 2)..” to “(Table 2).”

·      Line 479: change “will be” to “was”

·      Lines 480, 481, 485, 489, 497, 498, 500: change “is” to “was” since the paper reports an already completed study findings. Kindly check other sections and use the appropriate English tenses suitable for the journal where necessary.

·      Line 531: correct “6 Conclusion and Discussion” to “6 Discussion and Conclusion” and change the subsections of line 532: “6.1 Conclusion”, line 554: “6.2 further discussions”, line 582: “6.3 Theoretical Contributions”, and line 606: “6.4 Policy Suggestions” to “6.1 further discussions”, “6.2 Theoretical Contributions”, “6.3 Policy Suggestions, “6.4 Conclusion”.

·      Line 629: correct “WeChat, TikTok, Weibo, and Facebook” to “WeChat TM, TikTok TM, Weibo TM, and Facebook TM.”

Comments on the Quality of English Language

Minor corrections needed

Author Response

We are grateful to the editor and reviewers for their positive and constructive comments and criticisms concerning our manuscript, Valuable data "gain" and "loss": The quantitative impact of information choice on consumers' decision to buy agricultural products(ID: foods-3188873). These comments and criticisms are very helpful for revising and improving our paper. We have made necessary corrections and changes in response to them. We hope that you will find our revised manuscript is acceptable for publication. Of course, we will make additionally changes if there remain unaddressed or inadequately addressed comments.

Reviewer #1:

  1. Abstract: lines 19 and 21, 23 kindly avoid the use of “we”and use wording such as “the study”.

Thanks for your observation and suggestion. We have changed.

  1. Line 41: correct “an”to “a”.

We appreciate these valuable comments. We have changed.

  1. The study entirety centers on “selenium-rich agricultural products” and I wonder why the title isn’t incorporating “selenium-rich agricultural products” for coherence but still holding on to the wording “agricultural product” which to a great extent presents a wide scope of studies relative to the agriculture field but the study does not. The write-up first presents the lack of micronutrients as the integral cause of “hidden hunger” and further explicitly acknowledges the vital merits and demerits of selenium which is a micronutrient. Further, the paper details information on “selenium-rich agricultural products” coupled with genetic programming (biotechnology) to enhance crops with selenium. Lastly, lines 86-87 “However, there is still a lack of in-depth research on consumer food safety information for nutrition-enhancing products, especially selenium-rich agricultural products”. This statement indicates the study is centered on “selenium-rich agricultural products”. Authors should kindly revise the paper title and embed “selenium-rich agricultural products” to make the manuscript more coherent and scientific.

Thank you for pointing this out. The title has changed. “Valuable data "gain" and "loss": The quantitative impact of information choice on consumers' decision to buy selenium-rich agricultural products”

  1. Lines 136-141: is the statement indicating that the lack of information on selenium-rich agricultural products accessible to consumers influences the effect (negative or positive) of selenium-rich agricultural products consumption since line 44 states “inadequate or excessive selenium levels can cause physical discomfort”, Also, lines 54-56 states “Compared to commonly used nutritional supplements such as iodized salt, iron soy sauce, and  fortified rice, there is still a lack of awareness regarding the cognition and recognition of selenium-rich agricultural products” and line 57 states “Consumers tend to be skeptical about the nutritional value of selenium”. These statements seem to reveal that “selenium-rich agricultural products” producers reluctantly avoid providing nutritional detail on the selenium product for consumers perusal. If so, why? Is selenium presence in a product that bad to affect product sales should buyers identify it as an ingredient in a product? Again, is the skeptical nature of consumers about selenium-rich agricultural products the reason for the study? Authors should justify these sections for coherence and avoid any discrepancies among readers after publication.

We appreciate these valuable comments. We have changed in the manuscript.

In line 45: However, selenium is an indispensable trace element for the human body, and either insufficient or excessive selenium levels can give rise to physical discomfort [9]. Severe selenium deficiency is closely associated with the prevalence of endemic Keshan disease [10].

In line 60: In the absence of selenium public education, a significant number of consumers tend to hold skeptical attitudes towards the significance of selenium supplementation [14].

Further clarification: The supplementation of selenium is indispensable. However, the promotion of selenium-rich products has been impeded due to the inadequate public understanding and the scarcity of accessible information.

  1. Lines 161-162: “The majority of consumers are not experts, and they assess the food microbiological standards of products based on color and appearance.” This statement should be revised as “The majority of consumers are not experts, and they assess the food microbiological standards of products based on color and appearance which is wrong given food safety and quality”. This is because the color and appearance of food from a food science perspective don’t merit a food wholesome or microbiologically safe for consumption. Therefore, even if buyers or consumers consider these indicators of food to be “microbially safe” it could be pardoned based on their lack of expertise but to a great extent technically and scientifically it must be a “wrong” approach to judge a “safe and quality food”. Hence, the recommended revised statement could be considered and changes affected at various sections where necessary.

Thank you for pointing this out. We have changed in the manuscript.

  1. Line 209: the wording “state of flow” seems confusing. Kindly revise to a better word for clarity.

We appreciate these valuable comments. We have changed to “state of flow experience”.

  1. Lines 237-246: instead of H5, H5a, H5b, H5c, and H5d why not consider H5a, H5b, H5c, H5d and H5e.

We appreciate these valuable comments. H5a, H5b, H5c, and H5d are the decompositions of H5.

  1. Line 267: change “we” and “our” to “the study”

Thank you for pointing this out. We have changed in the manuscript.

  1. Line 342: there must be a space after the period in “(66.19%).A” to “(66.19%). A”

We appreciate these valuable comments. We have changed in the manuscript.

  1. Line 351: delete one of the periods “(Table 2)..” to “(Table 2).”

Thank you for pointing this out. We have changed in the manuscript.

  1. Line 479: change “will be” to “was”

Thank you for pointing this out. We have changed in the manuscript.

  1. Lines 480, 481, 485, 489, 497, 498, 500: change “is” to “was” since the paper reports an already completed study findings. Kindly check other sections and use the appropriate English tenses suitable for the journal where necessary.

Thank you for pointing this out. We have changed in the manuscript.

  1. Line 531: correct “6 Conclusion and Discussion” to “6 Discussion and Conclusion” and change the subsections of line 532: “6.1 Conclusion”, line 554: “6.2 further discussions”, line 582: “6.3 Theoretical Contributions”, and line 606: “6.4 Policy Suggestions” to “6.1 further discussions”, “6.2 Theoretical Contributions”, “6.3 Policy Suggestions, “6.4 Conclusion”.

Thank you for pointing this out. We have changed in the manuscript.

  1. Line 629: correct “WeChat, TikTok, Weibo, and Facebook” to “WeChat TM, TikTok TM, Weibo TM, and Facebook TM.”.

Thank you for pointing this out. We have changed in the manuscript.

Reviewer 2 Report

Comments and Suggestions for Authors

Minor detail, but on page 17 you should capitalize 6.2 further discussions.

Author Response

We are grateful to the editor and reviewers for their positive and constructive comments and criticisms concerning our manuscript, Valuable data "gain" and "loss": The quantitative impact of information choice on consumers' decision to buy agricultural products(ID: foods-3188873). These comments and criticisms are very helpful for revising and improving our paper. We have made necessary corrections and changes in response to them. We hope that you will find our revised manuscript is acceptable for publication. Of course, we will make additionally changes if there remain unaddressed or inadequately addressed comments.

Reviewer #2:

Minor detail, but on page 17 you should capitalize 6.2 further discussions.

Thanks for your observation and suggestion. We have changed.